# Associations of upper respiratory mucosa microbiota with Rheumatoid arthritis, autoantibodies, and disease activity

Young Bin Joo[1,2☉], Juho Lee[3,4☉], Yune-Jung Park[5], So-Young Bang[1,2], Kwangwoo Kim[3,4]*, Hye-Soon Lee[1,2]*

**1** Department of Internal Medicine, Division of Rheumatology, Hanyang University Guri Hospital, Guri, Republic of Korea, **2** Hanyang University Institute for Rheumatology Research, Seoul, Republic of Korea, **3** Department of Biology, Kyung Hee University, Seoul, Republic of Korea, **4** Department of Biomedical and Pharmaceutical Sciences, Kyung Hee University, Seoul, Republic of Korea, **5** Department of Internal Medicine, Division of Rheumatology, St. Vincent's Hospital, College of Medicine, The Catholic University of Korea, Seoul, Republic of Korea

☉ These authors contributed equally to this work.

* kkim@khu.ac.kr (KK); lhsberon@hanyang.ac.kr (HSL)

**Data Availability Statement:** Gene sequencing data used for this study are available at the NIH's

## Abstract

The lung is recognized as a site for initiating the formation of self-antigen and autoimmune responses in rheumatoid arthritis (RA). We aimed to investigate the association of upper respiratory microbiota with RA, autoantibody production, and disease activity. Forty-six patients with RA and 17 controls were examined. Nasopharyngeal swab samples were sequenced for microbiome profiling using the V3–V4 region of the 16S rRNA gene. The microbial diversity and relative abundance were compared between RA patients and controls. Correlation analyses were conducted to evaluate the relationship between microbial abundance and clinical markers such as autoantibodies and disease activity. Microbial diversity analysis revealed no major differences between RA patients and healthy controls. However, beta diversity analysis indicated a subtle distinction in microbial composition (unweighted UniFrac distance) between the two groups ($P = 0.03$), hinting at a minor subset of microbiota associated with disease status. Differential abundance analysis uncovered specific taxa at various taxonomic levels, including *Saccharibacteria* (TM7) [O-1] ($P_{FDR} = 2.53 \times 10^{-2}$), TM7 [F-1] ($P_{FDR} = 5.20 \times 10^{-3}$), *Microbacterium* ($P_{FDR} = 3.37 \times 10^{-4}$), and *Stenotrophomonas* ($P_{FDR} = 2.57 \times 10^{-3}$). The relative abundance of ten genera correlated significantly with anti-cyclic citrullinated peptide (anti-CCP) antibody levels ($P_{FDR} < 0.05$) and 11 genera were significantly associated with disease activity markers, including ESR, CRP, DAS28-ESR, and DAS-CRP ($P_{FDR} < 0.05$). In particular, *Saccharibacteria* TM7 [G-3] and *Peptostreptococcaceae* [XI] [G-1] were correlated with all disease activity biomarkers. Dysbiosis in the upper respiratory mucosa is associated with RA, anti-CCP antibody levels, and disease activity.

Sequence Read Archive (SRA) under accession number PRJNA1058141.

**Funding:** This research was supported by the research fund of Rheumatology Research Foundation (RRF-2020-02) (YBJ), St. Vincent's Hospital; the research institute of medical science (SVHR-2020-12) (YBJ); and the Basic Science Research Program through the National Research Foundation of Korea (NRF) funded by the Ministry of Education (NRF-2021R1A6A1A03038899) (HSL).

**Competing interests:** The authors have declared that no competing interests exist.

## Introduction

Rheumatoid arthritis (RA) is a chronic autoimmune disorder that primarily affects the joints, leading to persistent inflammation, joint damage, and loss of function. In addition to its impact on the joints, RA can also lead to extraarticular manifestations in various organs, including the lungs, heart, and skin [1].

In patients with RA, the lungs play a pivotal role in two clinical aspects. First, they serve as the site where extraarticular manifestations, such as interstitial lung disease and bronchiectasis, occur [2, 3]. Second, they are associated with the initiation of immune responses linked to RA, which can ultimately result in synovitis, a key pathogenesis of RA [4, 5]. The proteins in lung tissue can undergo citrullination that can be influenced by environmental factors such as smoking. The citrullinated proteins have a high affinity for binding to major histocompatibility complex molecules, especially those encoded by *HLA-DRB1* with RA-risk variants. In genetically susceptible individuals, this interaction may activate T cells and stimulate the maturation of B cells, ultimately resulting in the production of autoantibodies specific to RA, known as anti-citrullinated protein antibodies (ACPA) [4].

Previous studies have supported the importance of the lungs in initiating immune responses associated with RA. For example, RA-associated autoantibodies have been detected in the sputum of individuals at risk of developing RA, as well as in those with early RA [6]. Moreover, immune cell activation has been observed in both the bronchoalveolar lavage (BAL) and bronchial tissue of patients with untreated RA without any lung diseases [6]. Furthermore, microbiome studies using 16S rRNA gene sequencing have revealed dysbiosis in the BAL fluid of early patients with RA who have not received disease-modifying antirheumatic drugs (DMARDs) compared to healthy controls [7]. Furthermore, another microbial study that examined patients with granulomatosis polyangiitis (GPA), using patients with RA serving as the disease control group, reported that some bacteria exhibited altered abundance levels that were shared between GPA and RA when compared to the healthy control group [8]. These findings suggest a potential association between the respiratory microenvironment and RA.

Based on the research findings above, we hypothesized that alterations in the microbial community of the upper respiratory mucosa may be associated with RA and its clinical manifestations. Therefore, in this study, we aimed to characterize the composition of the microbiota in the upper respiratory mucosa of patients with RA compared to healthy individuals using 16S rRNA gene sequencing. We also investigated the associations of the upper respiratory microbiota with RA-related autoantibodies and disease activity.

## Materials and methods

### Study design and participants

This study was a prospective cross-sectional study, which included patients with RA and healthy controls recruited from September 9, 2020, to February 9, 2021 from the Department of Rheumatology at St. Vincent's Catholic University Hospital in South Korea. The case group consisted of patients with RA aged between 19 and 74 years, all of whom satisfied the 2010 European League Against Rheumatism-American College of Rheumatology criteria for RA [9]. The control group comprised age-matched healthy participants who exhibited no signs of arthralgia or arthritis. In addition, the control group had no prior history of autoimmune diseases (including RA), no chronic respiratory diseases (e.g., interstitial lung disease or chronic obstructive pulmonary disease), and no family history of autoimmune diseases. Participants were excluded from either group if they had taken antibiotics within the last 3 months, suffered

from upper or lower respiratory tract symptoms within the last 3 months, or had received a cancer diagnosis within the previous 5 years. This study was approved by the Institutional Ethics Review Board of St. Vincent's Catholic University Hospital (VC20TASI0147). All research methods were performed in accordance with relevant guidelines and regulations. All patients provided written informed consent prior to participation.

## Clinical data

Clinical data were collected from all participants during routine clinical practice, including age at the time of nasal sampling, sex, smoking history (categorized as never and ever/current smokers), and comorbidities such as diabetes mellitus, hypertension, viral hepatitis, and dyslipidemia. The clinical data collected from the patients with RA included rheumatoid factor (RF), anti-cyclic citrullinated peptide (anti-CCP) antibody, disease activity score 28 (DAS28)-erythrocyte sedimentation rate (ESR), DAS28-C-reactive protein (CRP), and medications taken within the last 3 months, including nonsteroid anti-inflammatory drugs, steroids, conventional synthetic disease-modifying anti-rheumatic drugs (DMARDs), and biologic DMARDs. Anti-CCP antibody was analyzed by chemiImmunoassay (Abbott Laboratories, IL, USA), and a positive reading was defined with a cutoff value of 5 U/mL. The antibody concentration maximum was defined as 340 U/mL, and for statistical purposes, the value of 340 U/mL was assigned to all measurements > 340 U/mL. RF titers were measured with a latex agglutination test (Beckman Coulter, CA, USA) with a cutoff value of 14 U/mL.

## 16S rRNA amplicon sequencing analysis and decontaminant sequence removal

Nasopharyngeal mucosa samples were obtained by swabbing the middle meatus with sterile specimen collection swabs, before storing at −80˚C. After assessing the quality of the isolated DNA, we generated the amplicons of the V3 and V4 regions of 16S rRNA genes using Herculase II Fusion DNA Polymerase and an Illumina Nextera XT library preparation kit v2, following the manufacturer's recommended protocols for library preparation. Subsequently, the indexed sequencing libraries were subjected to paired-end sequencing using Illumina MiSeq sequencers. The sequencing data generated in this study are available at the NIH's Sequence Read Archive (SRA) under accession number PRJNA1058141.

The sequenced reads were processed using the QIIME2 platform [10]. Index trimming and denoising steps were conducted via the DADA2 plug-in [11]. An amplicon sequence variant (ASV) method was employed in the denoising step to correct erroneous sequences, including chimeric reads. The expanded Human Oral Microbiome Database (eHOMD, RefSeq v15.22) [12] was used as a reference for taxonomic classification and phylogenetic annotation.

To account for the potential presence of contaminant bacteria originating from external sources during sample preparation, we excluded known contaminant taxa that had never been identified as oral bacteria based on existing oral and contamination databases [13]. The three contaminant genera, *Acidovorax*, *Roseomonas*, and *Serratia*, were excluded from all downstream analyses.

## Statistical analysis

To ensure the sensitivity of the diversity analysis, we used the rarefaction method to correct for different sequencing readout sizes of microbial data across samples before conducting diversity analysis. The alpha diversity of the upper respiratory microbiota was quantified using species richness, the Shannon index, and the Gini–Simpson index. Statistical differences in alpha diversity between the two groups were determined using the Mann–Whitney U test. The beta

diversity, for examining differences in microbial composition among individual samples, was analyzed using principal coordinate analysis (PCoA) based on two distance metrics: the unique fraction (UniFrac) and Bray–Curtis dissimilarity. We assessed statistical differences in beta diversity between the sample groups using distance-based permutational multivariate analysis of variance (PERMANOVA) [14].

Differential abundance analysis of individual taxa between sample groups of interest (e.g., disease status, autoantibody, and disease activity markers) was performed using the DESeq2 software (*version*: 1.32.0) [15]. This analysis estimated $\log_2$-fold changes in normalized ASV counts, adjusting for both sex and age. The Benjamini-Hochberg procedure was applied to provide FDR-corrected p-values.

Pearson correlation coefficients were calculated to explore the potential association between the abundance of upper respiratory mucosa-associated taxa and various RA-related characteristics, such as disease activity markers and autoantibody titers among RA patients.

We compared the baseline characteristics of the RA patients and controls using an independent Student's t-test for continuous variables and using Fisher's Exact test for categorical variables. All statistical analyses were performed using R (version 4.1.0).

## Results

### Baseline characteristics of the participants

We recruited 46 patients with RA and 17 age-matched healthy controls (mean age ± standard deviation: 57.0 ± 8.1 vs. 56.8 ± 10.9 years, RA vs. control, $P$ = 0.958). There was no significant difference in sex distribution between RA cases and healthy controls (female: 78.3% vs. 94.1%, RA vs. control, $P$ = 0.262; Table 1). Among the patients with RA, 65.2% (30/46) had never smoked, whereas all controls were never-smokers. There were no significant differences in comorbidities, including diabetes mellitus, hypertension, viral hepatitis, and dyslipidemia, between the two groups. Among the 45 RA cases from whom additional clinical information beyond age, sex, smoking status, and comorbidities was available, the mean disease duration was 9.2 ± 7.6 years. The mean levels of ESR and CRP were 22.4 ± 17.1 mm/h and 0.4 ± 0.7 mg/dl, respectively. Approximately 91.3% of patients with RA had been treated with conventional synthetic DMARDs, while 32.6% had been treated with biologic DMARDs.

### Bacterial diversity in the upper respiratory mucosa of patients with RA and healthy controls

To assess the overall composition of the upper respiratory microbiota, which closely resembles that of the upper respiratory tract [16], we estimated the alpha and beta diversity at the genus level and compared them between patients with RA and controls. Alpha diversity, as measured by species richness, the Shannon diversity index, and the Gini–Simpson index, showed no significant differences between the two groups ($P$ > 0.05; Fig 1A–1C).

Beta diversity was assessed using the Bray–Curtis distance and both the weighted (quantitative) and unweighted (qualitative) UniFrac distances of the ASV count profiles between samples, and was visualized through PCoA plots (Fig 2A–2C).

The analysis indicated a slight difference in microbial composition between patients with RA and healthy controls based on the unweighted UniFrac distance ($P$ = 0.03, Fig 2A). However, no significant differences were observed in the weighted UniFrac distance ($P$ = 0.119, Fig 2B) or the Bray–Curtis distance ($P$ = 0.094, Fig 2C) between the two groups. These findings indicate that microbial communities in the upper respiratory mucosa of patients with RA and healthy controls largely share microbiota in terms of microbial composition and abundance.

**Table 1. Characteristics of the study population at the time of nasal sampling.**

| | RA case (n = 46) | Healthy control (n = 17) | P |
|---|---|---|---|
| Age at nasal sampling, years | 57.0 ± 8.1 | 56.8 ± 10.9 | 0.958 |
| Sex, female | 36 (78.3) | 16 (94.1) | 0.262 |
| Smoking | | | |
| Never | 30 (65.2) | 17 (100) | 0.003 |
| Ever or current | 16 (34.8) | 0 | |
| Comorbidities | | | |
| Diabetes mellitus | 11 (24.4) | 4 (23.5) | 1.000 |
| Hypertension | 13 (28.9) | 7 (41.2) | 0.356 |
| Viral hepatitis | 3 (6.8) | 0 | 0.553 |
| Dyslipidemia | 6 (13.3) | 1 (5.9) | 0.662 |
| RA-related characteristics | RA case (n = 45[a]) | | |
| Age at RA diagnosis | 48.3 ± 9.1 | | |
| Age at nasal sampling, years | 56.8 ± 8.1 | | |
| Disease duration | 9.2 ± 7.6 | | |
| RF positivity | 38 (82.6) | | |
| RF level | 96.1 ± 97.9 | | |
| Anti-CCP antibody positivity | 42 (91.3) | | |
| Anti-CCP antibody level | 131.9 ± 77.5 | | |
| ESR, mm/h | 22.4 ± 17.1 | | |
| CRP, mg/dl | 0.4 ± 0.7 | | |
| DAS28 (ESR) | 2.5 ± 0.9 | | |
| DAS28 (CRP) | 2.0 ± 0.7 | | |
| Radiographic damage | | | |
| Erosion on hands or feet | 19 (41.3) | | |
| Joint space narrowing on hands or feet | 22 (47.8) | | |
| Medication | | | |
| NSAIDs | 30 (65.2) | | |
| Steroid | 34 (73.9) | | |
| csDMARDs | 42 (91.3) | | |
| bDMARDs | 15 (32.6) | | |

[a]One among the 46 cases was excluded in the analysis due to limited clinical information.

Data were presented as means with standard deviations or as number with percentages.

RA, Rheumatoid arthritis; RF, Rheumatoid factor; CCP, Cyclic citrullinated peptide; ESR, Erythrocyte sedimentation rate; CRP, C-reactive protein; DAS, Disease activity score; NSAIDs, Non-steroid anti-inflammatory drugs; csDMARDs, Conventional synthetic disease modifying antirheumatic drugs; bDMARDs, Biologic disease modifying antirheumatic drugs.

Nevertheless, the beta diversity results may imply the presence of a distinct minor subset of microbiota exhibiting differential abundance levels according to disease status.

## Identification of differentially abundant taxa in the upper respiratory mucosa microbiota of patients with RA

The upper respiratory mucosa of patients with RA exhibited prominent bacterial phyla, including *Firmicutes*, *Actinobacteria*, and *Proteobacteria*, which aligned with the bacterial composition found in healthy controls. Similarly, at the genus level, several genera, including

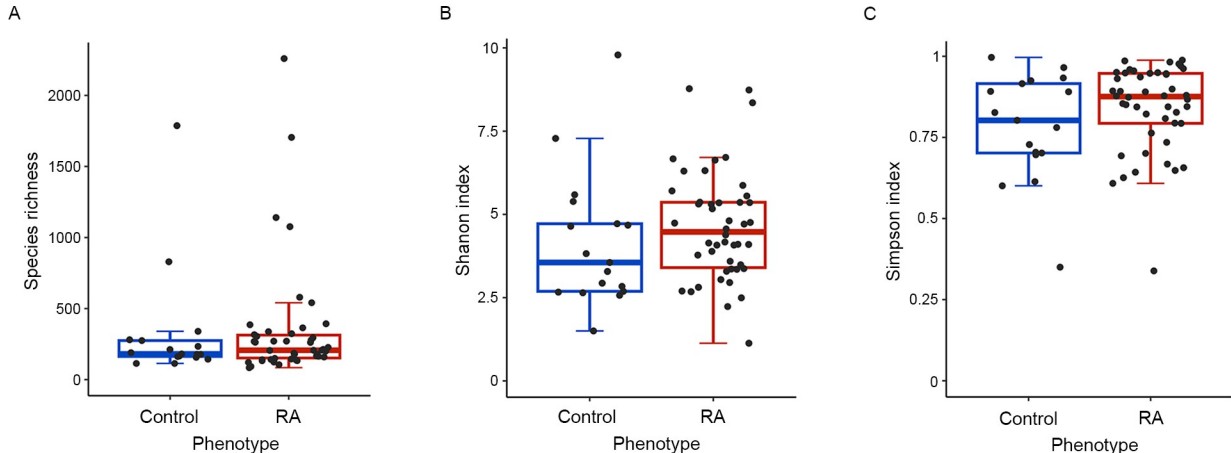

**Fig 1. Comparison of alpha diversity in upper respiratory mucosa microbiota between patients with RA and healthy controls.** Alpha diversity was assessed using species richness (A), the Shannon index (B), and the Simpson index (C). Statistical differences in alpha diversity between the two groups were determined using the Mann–Whitney U test. RA: Rheumatoid arthritis.

*Corynebacterium* and *Staphylococcus*, were highly abundant within the upper respiratory mucosa and consistently the most abundant in both patients with RA and healthy controls (Fig 3).

We next conducted differential abundance analysis, which revealed significant differences in several individual taxa between the RA and healthy controls (Fig 4).

Specifically, at the genus level, unclassified *Saccharibacteria* (TM7) [F-1] and unclassified *Enterobacterales* showed the most significant abundance differences between RA and healthy controls after adjusting for multiple comparisons ($P_{FDR} = 1.91 \times 10^{-4}$ and $1.90 \times 10^{-5}$, respectively). In addition, four other genera (unclassified *Sphingomonadaceae*, *Microbacterium*, *Stenotrophomonas*, and unclassified *Proteobacteria*) exhibited higher abundance in patients with RA ($P_{FDR} < 0.05$), whereas the genus unclassified bacilli was significantly less abundant in patients with RA compared to healthy controls ($P_{FDR} = 0.019$).

The same analysis, conducted at the family and order levels, identified significant differences in the abundance levels of seven families and four orders between the RA and healthy

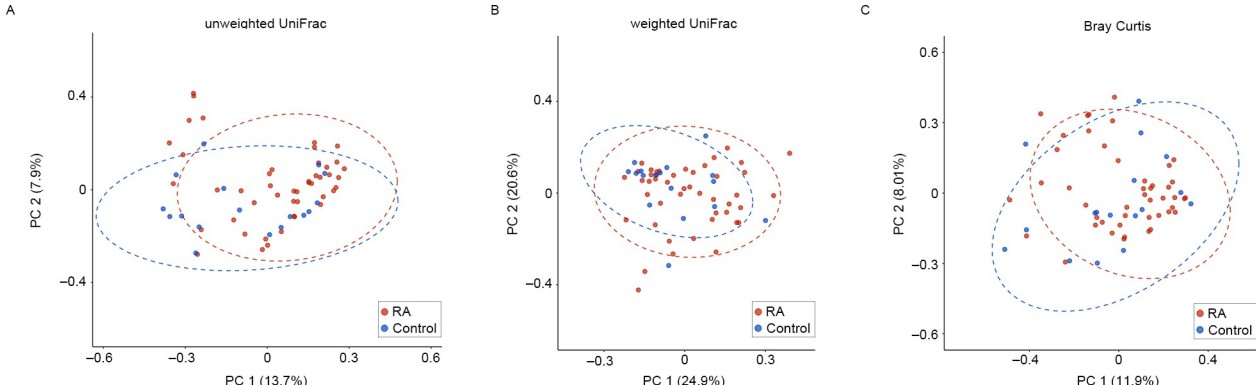

**Fig 2. Comparison of beta diversity in upper respiratory mucosa microbiota between patients with RA and healthy controls.** Beta diversity was estimated and visualized on the top two PCoA axes based on the UniFrac distance (A: unweighted; B: weighted) and Bray-Curtis distance (C). Statistical differences in beta diversity between the two groups were determined using PERMANOVA. Abbreviations: RA, Rheumatoid arthritis; PCoA, Principal coordinate analysis; PERMANOVA, Permutational multivariate analysis of variance.

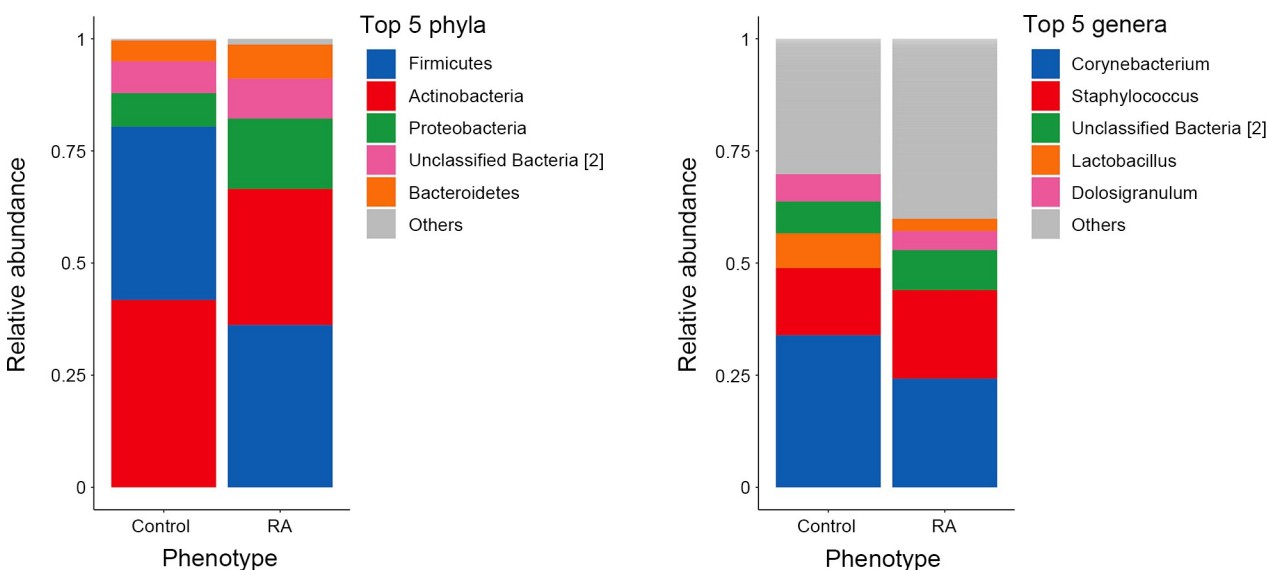

**Fig 3. Taxonomic summaries for the most abundant taxa at the phylum and genus levels.** The stacked and colored bars represent the relative abundance of the top five most abundant taxa in the upper respiratory mucosa of the study subjects. The other phyla or genera are categoized as "Others" and are shown in gray. Unclassified taxa with different ASVs but the same taxonomic name are distinguished by different numbers in square brackets. Abbreviations: RA, Rheumatoid arthritis; ASV, Amplicon sequence variant.

control groups (Fig 4). Notably, the higher taxonomic ranks, to which the most significant RA-specific genus *Saccharibacteria* belongs, consistently exhibited higher abundance levels in RA, both at the family ($P_{FDR} = 5.20 \times 10^{-3}$) and order levels ($P_{FDR} = 2.53 \times 10^{-2}$), as well as at the genus level ($P_{FDR} = 1.91 \times 10^{-4}$), with large fold changes in abundance.

Because some patients with RA had a history of cigarette smoking, whereas healthy controls did not, we performed additional stratification analyses to demonstrate that the identified disease-specific microbial changes were not influenced by the smoking behavior of patients. As a result, we observed highly consistent diversity indices between patients with RA who had never smoked (n = 30) and those who had (n = 16). Furthermore, when we performed a differential abundance analysis for individual taxa using only non-smoker patients and controls, the results were highly consistent with those obtained using all patients and controls (S1 Fig).

## Correlations between the relative abundance of taxa and RA-related characteristics

We next conducted correlation analyses to investigate potential associations between the abundance of upper respiratory mucosa taxa and autoantibodies, such as anti-CCP antibodies, and RF, as well as disease activity indices. Our findings revealed that ten genera were significantly associated with the titer of anti-CCP antibodies, which are thought to be initially produced in the respiratory microenvironment (Fig 5).

The relative abundance of seven out of these ten genera significantly decreased as the titer of anti-CCP antibody increased. As for RF, three genera showed significant correlations with its titers.

Microbial associations with anti-CCP antibody and RF were generally distinct, except that the genus *Mollicutes* [G-2] exhibited a significantly negative correlation with both autoantibodies. The distinct association patterns and the larger number of anti-CCP antibody-associated taxa with larger abundance fold changes suggest that the levels of anti-CCP antibody and RF are not associated with the same microbiota-mediated etiology in the upper respiratory

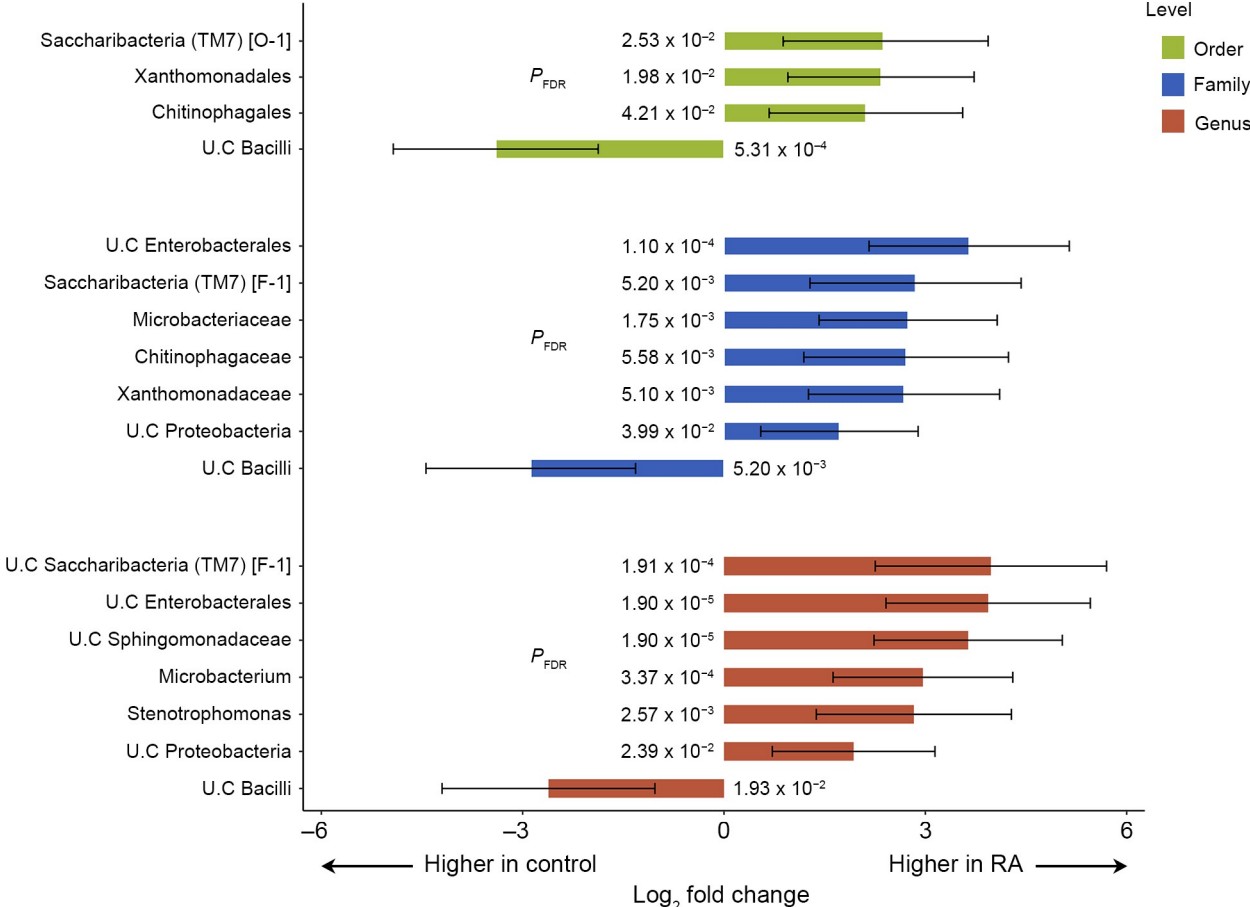

**Fig 4. Differential abundance analysis of individual microbial taxa in patients with RA compared to healthy controls.** The bar plot illustrates the significantly differential abundance changes in individual microbial taxa at the order, family, and genus taxonomic levels. Each bar represents a log2-fold change in the mean abundance of a taxon in patients with RA compared to healthy controls, with error bars corresponding to the 95% confidence intervals. BH FDR-corrected P-values for the log2-fold change are provided. Abbreviations: RA, Rheumatoid arthritis; BH FDR, Benjamini-Hochberg false discovery rate.

mucosa and that the production of anti-CCP antibody is more influenced by respiratory microbiota than RF.

Subsequently, we conducted a correlation analysis between disease activity and the relative abundance of taxa. To estimate the disease activity, we employed four well-established indices: ESR, CRP, DAS28-ESR, and DAS28-CRP. We identified a set of genera that exhibited significantly positive or negative correlations between their relative abundance and the degree of disease activity. Notably, three genera, including Peptostreptococcaceae [XI] [G-1], *Johnsonella*, and *Saccharibacteria* (TM7) [G-3], consistently showed significantly positive associations with all four disease activity indices. In particular, *Saccharibacteria* (TM7), identified as the significant RA-risk taxa at family and order levels in our case-control study, was identified as a genus that was positively correlated with disease activity in case-only analysis.

## Discussion

The respiratory mucosa has been suggested to play a role in initiating autoimmune responses in patients with RA, including the production of citrullinated autoantigens. In this study, we report the following key findings regarding the microbiota of the upper respiratory mucosa in

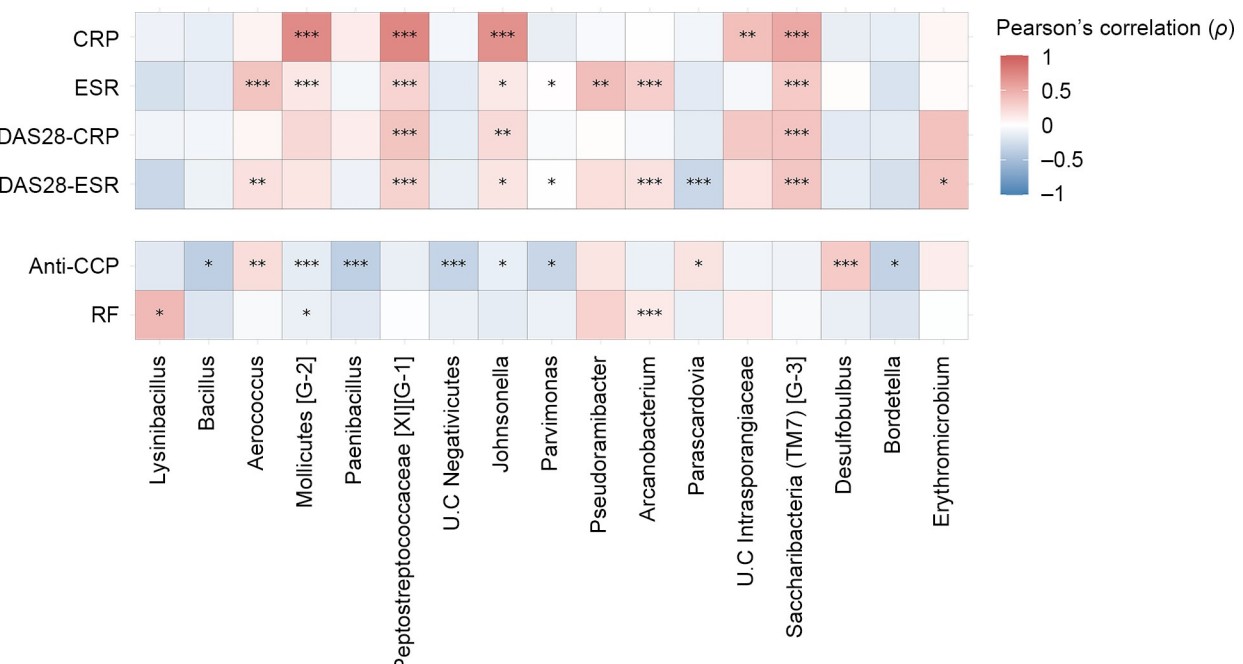

**Fig 5. Correlation between upper respiratory mucosa microbiota and clinical markers in patients with RA.** Pearson correlation coefficients were calculated to examine the relationship between the abundance level of individual genera and disease activity markers or autoantibody titers in patients with RA. The genera that showed a significant linear relationship with at least one clinical factor were included in the heatmap of correlation coefficients (* for PFDR < 0.05, ** for PFDR < 0.01, *** for PFDR < 0.001). Abbreviations: CRP, C-reactive protein; ESR, Erythrocyte sedimentation rate; DAS, Disease activity score; anti-CCP, Anti-cyclic citrullinated peptide antibody; RF, Rheumatoid factor.

patients with RA. First, we identified differences in the relative abundance of several taxa between patients with RA and the control group. Second, we found an association between microbial abundance and the titer of anti-CCP antibody and disease activity indicators.

Despite the well-known role of the lung as an extraarticular mucosal site where RA immune responses begin, microbiome research has predominantly focused on the gut mucosa [17–19]. Microbiome research on the lung is challenging because of the invasive approach needed to access BAL fluid or lung tissue samples, as well as the limited biomass of these samples. However, a previous study demonstrated that the bacterial composition of the lung is indistinguishable from that of the upper respiratory tract in healthy subjects [16]. Thus, we conducted 16S rRNA sequencing analysis using a less invasive nasopharyngeal swab rather than BAL fluid.

Although we did not detect notably large differences in diversity between the RA and control groups, there was marginal significance, indicating compositional differences in respiratory microbiota between the two groups based on the UniFrac distance. This result suggested the presence of a small fraction of microbiota associated with RA, prompting us to perform a differential abundance analysis for each taxon. Similar to our results, a previous study that used BAL fluid samples from early DMARD-naïve patients with RA reported that the overall structure of microbiota in patients with RA differed significantly from that of healthy controls based on the unweighted UniFrac distance [7]. In contrast, Lamprecht *et al.* found no significant differences in beta diversity between patients with long-standing RA on DMARDs and the control group when using nasopharyngeal samples [8]. This discrepancy may be explained by different clinical characteristics, such as disease duration and prescribed medication, as well as variations in genetic and environmental factors. Both our study and that of Lamprecht *et al.* [8] included patients who had mostly been treated with DMARDs and had similar disease durations. It is plausible that, as inflammation in RA resolves with DMARD treatment,

intrinsic RA-specific microbial composition and abundance levels may gradually diminish, becoming less distinct compared to the control group. Indeed, a previous metagenomic shotgun sequencing analysis demonstrated that RA-associated dysbiosis in the gut and oral cavity was partially resolved after treatment [20]. A similar finding was reported in a study of patients with GPA using nasal swab samples, where patients with GPA who were not on immunosuppressive therapy had significantly different beta diversities compared to the controls, whereas those on immunosuppressive therapies were similar to controls [21].

In this study, we observed a significantly differential abundance of several respiratory microbiota in patients with RA compared to healthy controls, particularly an increased presence of potentially harmful bacteria. Among the detected bacteria, we reported for the first time that *Saccharibacteria* (TM7) in the upper respiratory mucosa exhibited higher abundance in patients with RA at several taxonomic levels, with strong statistical significance. Higher abundance levels of the phylum *Saccharibacteria* (TM7) have been reported to be associated with chronic inflammation [22] and periodontitis [23–25], both of which are risk factors for RA development [26]. Moreover, a higher abundance of the phylum *Saccharibacteria* (TM7) has also been detected in the saliva of patients with juvenile idiopathic arthritis compared to that in healthy controls [27]. In this study, we observed an 8.2-fold increase in the abundance of the phylum *Saccharibacteria* (TM7) in patients with RA compared to healthy controls ($P_{FDR}$ = $1.33 \times 10^{-3}$, S2 Fig). In addition, the order *Saccharibacteria* (TM7) [O-1] and the family *Saccharibacteria* (TM7) [F-1] on respiratory mucosa showed higher abundance levels in patients with RA than in healthy controls. Furthermore, the abundance of the genus *Saccharibacteria* (TM7) [G-3] was significantly and positively correlated with the level of disease activity markers, including ESR, CRP, DAS28-ESR, and DAS28-CRP, suggesting that *Saccharibacteria* (TM7) contributes to chronic inflammation in RA.

In the correlation analysis between the relative abundance of upper respiratory mucosal microbiota and RA-associated clinical markers, we revealed two important findings. First, the anti-CCP antibody showed a larger number of significant microbial associations than RF, implying that the respiratory microbiota has a more notable impact on anti-CCP antibodies than on RF. Unlike RF, anti-CCP antibodies are initially produced in the lungs after exposure to triggering factors such as smoking and environmental pollution [28, 29]. Our findings suggest that dysbiosis in the upper respiratory mucosa triggers the production of anti-CCP antibodies, although further research is needed to understand the causal association between them.

Second, we identified several microbial taxa that were associated with disease activity but were distinct from those associated with anti-CCP antibody. It could be hypothesized that dysbiosis in the upper respiratory mucosa, associated with changes in the abundance of specific taxa, leads to the production of anti-CCP antibodies, and the abundance changes in another set of specific bacteria may subsequently play a relevant role in the onset of systemic inflammation in patients with RA. Nevertheless, these findings and hypotheses should be interpreted with caution due to the absence of longitudinal human studies in preclinical and early DMARD-naïve patients with RA, as well as the lack of mechanism analysis.

This study has several limitations that warrant discussion. First, we could not determine whether the observed dysbiosis was a trigger in the development of RA or a secondary phenomenon following RA onset. Second, we found significant associations between abundance changes in several microbial taxa at the order, family, and genus taxonomic levels. However, at the genus level, several taxa were unclassified, except for the genera *Stenotrophomonas* and *Microbacterium*. Conducting further studies using larger sample sizes and metagenomic shotgun sequencing could offer more comprehensive insights into the roles and mechanisms of dysbiosis at respiratory sites in RA.

## Conclusions

We observed differences in the relative abundance of several bacteria in the upper respiratory mucosa between patients with RA and healthy controls. We also identified significant correlations between the relative abundance levels of individual microbial taxa and the anti-CCP antibody titer, as well as the degree of disease activity. These findings emphasize the significance of the microbial community in respiratory organs in RA and provide potential therapeutic strategies for RA.

## Supporting information

**S1 Fig. Additional microbial analysis of subgroups to investigate the impact of smoking status.** Alpha (A) and beta (B) diversity were compared between RA patients who had never smoked and those who had. Alpha diversity was evaluated using the Shannon index, while beta diversity was assessed using the Bray-Curtis dissimilarity. Differential abundance analyses of microbial composition were conducted between never smokers among RA patients and controls (C). Each bar in the figures represents the relative abundance of differentially abundant microbiota, along with the 95% confidence interval, and the colors indicate distinct taxonomic levels. The *P* value was adjusted for multiple testing using the Benjamini-Hochberg method. RA; rheumatoid arthritis, PC; Principal Coordinate, U.C; unclassified.
(TIF)

**S2 Fig. Association of Saccharibacteria across taxonomic levels.** Each bar represents the relative abundance of differentially abundant Saccharibacteria, with 95% confidence intervals. Distinct taxonomic levels of Saccharibacteria are indicated by different colors. The *P* value was adjusted for multiple testing using the Benjamini-Hochberg method. RA; rheumatoid arthritis, U.C; unclassified.
(TIF)

## Author Contributions

**Conceptualization:** Young Bin Joo, Kwangwoo Kim.

**Data curation:** Young Bin Joo, Yune-Jung Park.

**Formal analysis:** Young Bin Joo, Juho Lee, Kwangwoo Kim.

**Funding acquisition:** Young Bin Joo.

**Investigation:** Yune-Jung Park.

**Methodology:** Juho Lee, So-Young Bang.

**Supervision:** Hye-Soon Lee.

**Writing – original draft:** Young Bin Joo, Juho Lee, So-Young Bang.

**Writing – review & editing:** Young Bin Joo, Kwangwoo Kim, Hye-Soon Lee.

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
