## [Decision Letter · Decision Letter 0]

21 May 2024

PONE-D-24-15432Associations of upper respiratory mucosa microbiota with rheumatoid arthritis, autoantibodies, and disease activityPLOS ONE

Dear Dr. Lee,

Thank you for submitting your manuscript to PLOS ONE. After careful consideration, we feel that it has merit but does not fully meet PLOS ONE’s publication criteria as it currently stands. Therefore, we invite you to submit a revised version of the manuscript that addresses the points raised during the review process.

We look forward to receiving your revised manuscript.

Kind regards,

Farah Al-Marzooq, MD, PhD

Academic Editor

PLOS ONE

Journal Requirements:

3. Thank you for stating the following financial disclosure: "This research was supported by the Research Foundation of the Korean College of Rheumatology (YBJ), St. Vincent’s Hospital, the research institute of medical science (SVHR-2020-12) (YBJ), and the Basic Science Research Program through the National Research Foundation of Korea (NRF) funded by the Ministry of Education (NRF-2021R1A6A1A03038899) (HSL)." 

Reviewers' comments:

Reviewer's Responses to Questions

**Comments to the Author**

1. Is the manuscript technically sound, and do the data support the conclusions?

Reviewer #1: Partly

2. Has the statistical analysis been performed appropriately and rigorously? 

Reviewer #1: No

3. Have the authors made all data underlying the findings in their manuscript fully available?

Reviewer #1: No

4. Is the manuscript presented in an intelligible fashion and written in standard English?

Reviewer #1: Yes

5. Review Comments to the Author

Reviewer #1: The study, titled 'Associations of upper respiratory mucosa microbiota with rheumatoid arthritis, autoantibodies, and disease activity,' aimed to investigate the association of upper respiratory microbiota with RA, autoantibody production, and disease activity. The study was conducted properly in general, and the writing is good. However, there are a few areas that need attention.

Major comments:

1. It’s not clear how the statistical analyses were conducted for the analyses of differential abundance, correlations, and baseline characteristics of the participants. The authors should provide a detailed description of the statistical analyses in the methods section.

2. Gene sequencing data used for this study are available at the NIH’s Sequence Read Archive (SRA) under accession number RJNA1058141. However, RJNA1058141 is not publicly available yet. The authors are required to make it publicly available before publication.

Minor comments:

1. In line 152, the citation format for ‘ref 14’ is wrong. The authors are advised to review the formatting of all references before publication.

6. PLOS authors have the option to publish the peer review history of their article (what does this mean?). If published, this will include your full peer review and any attached files.

Reviewer #1: No

---

## [Author Response · Author response to Decision Letter 0]

8 Jun 2024

I have attached the responses to the reviewer comments and journal requirements under the title 'Response to Reviewer'. I would appreciate it if you could check the attached file.

---

## [Editor Report · Decision Letter 1]

16 Jul 2024

Associations of upper respiratory mucosa microbiota with rheumatoid arthritis, autoantibodies, and disease activity

PONE-D-24-15432R1

Dear Dr. Lee,

We’re pleased to inform you that your manuscript has been judged scientifically suitable for publication and will be formally accepted for publication once it meets all outstanding technical requirements.

Kind regards,

Farah Al-Marzooq, MD, PhD

Academic Editor

PLOS ONE
---

## [Editor Report · Acceptance letter]

25 Jul 2024

PONE-D-24-15432R1 

PLOS ONE

Dear Dr. Lee, 

I'm pleased to inform you that your manuscript has been deemed suitable for publication in PLOS ONE. Congratulations! Your manuscript is now being handed over to our production team.

Kind regards, 

on behalf of

Dr. Farah Al-Marzooq 

Academic Editor

PLOS ONE